# Microsystem Nodes for Soil Monitoring via an Energy Mapping Network: A Proof-of-Concept Preliminary Study

**DOI:** 10.3390/mi13091440

**Published:** 2022-09-01

**Authors:** Anna Sabatini, Alfiero Leoni, Gil Goncalves, Alessandro Zompanti, Marco V. Marchetta, Paulo Cardoso, Simone Grasso, Maria Vittoria Di Loreto, Francesco Lodato, Costanza Cenerini, Etelvina Figuera, Giorgio Pennazza, Giuseppe Ferri, Vincenzo Stornelli, Marco Santonico

**Affiliations:** 1Unit of Computational Systems and Bioinformatics, Department of Engineering, Campus Bio-Medico University of Rome, 00128 Rome, Italy; 2Department of Industrial and Information Engineering, University of L’Aquila, 67100 L’Aquila, Italy; 3Centre for Mechanical Technology and Automation, Department of Mechanical Engineering, University of Aveiro, 3810-193 Aveiro, Portugal; 4Unit of Electronics for Sensor Systems, Department of Engineering, Università Campus Bio-Medico di Roma, 00128 Rome, Italy; 5Department of Biology & CESAM, University of Aveiro, 3810-193 Aveiro, Portugal; 6Unit of Electronics for Sensor Systems, Department of Science and Technology for Humans and the Environment, Campus Bio-Medico University of Rome, 00128 Rome, Italy

**Keywords:** soil monitoring, voltammetric sensors, microbial fuel cell, precision agriculture

## Abstract

The need for accurate information and the availability of novel tool and technological advances in agriculture have given rise to innovative autonomous systems. The aim is to monitor key parameters for optimal water and fertilizer management. A key issue in precision agriculture is the in situ monitoring of soil macronutrients. Here, a proof-of-concept study was conducted that tested two types of sensors capable of capturing both the electrochemical response of the soil and the electrical potential generated by the interaction between the soil and plants. These two sensors can be used to monitor large areas using a network approach, due to their small size and low power consumption. The voltammetric sensor (BIONOTE-L) proved to be able to characterize different soil samples. It was able, indeed, to provide a reproducible voltammetric fingerprint specific for each soil type, and to monitor the concentration of CaCl_2_ and NaCl in the soil. BIONOTE-L can be coupled to a device capable of capturing the energy produced by interactions between plants and soil. As a consequence, the functionality of the microsystem node when applied in a large-area monitoring network can be extended. Additional calibrations will be performed to fully characterize the instrument node, to implement the network, and to specialize it for a particular application in the field.

## 1. Introduction

Global warming, the reduction in the world’s arable land caused by drought and erosion, and the expectations for more sustainable practices to reduce the use of water and plant protection products, are becoming impellent requests for innovative solutions in the agri-food sector [1]. ‘Green’ options, such as solar power or biofuel, compete against other ways of using the land, such as food production. Since world population is increasing and arable land is decreasing, using sustainable alternatives such as precision agriculture (PA) can be an effective response.

Sustainable and precision agriculture (SA and PA) can be defined as an integrated system of practices and technologies applied for crop production, with in situ applications and management of natural resources on which the agricultural economy depends [2].

SA has the aim of improving environmental quality. In fact, it promotes the more efficient use of non-renewable resources, integrates respect for the natural biological cycles of plants, supports the economic viability of agricultural operations, and improves the quality of life of farmers and society as a whole [3]. Technologies that enable PA are recognized as one of the key ways to achieve these goals through the measuring, monitoring, and responding to the specific in situ variability of agricultural land.

PA requires a data acquisition system that allows the farmer to evaluate the heterogeneity of the physical and chemical content of the soil. Geospatial variations and crop data acquired in the field are also necessary to optimize the use of resources [4]. Increasingly accurate information and technological advancements have given rise to innovative autonomous systems, which have as their objectives the monitoring of humidity, pH, salinity, and soil organic matter and its structure [5,6,7]. Through monitoring these parameters, it is possible to optimally manage the use of resources in terms of water and fertilizers, thus addressing a paramount topic in PA: in situ monitoring of soil nutrients.

Although macronutrients exist in nature in many forms in the soil ecosystem, they are usually artificially supplied in the form of fertilizers [8]. In fact, when these macronutrients are present in limited quantities, plants respond by reducing their growth and modifying their morphology and physiology to maximize and acquire the few resources available. On the contrary, the nutrient input into soils by inorganic fertilization is only partially taken up by plants, while some remains in the soil or is leached into surface and underground aquifers during rain and irrigation. Surface waters are those destined for public and private water supplies, including agricultural ones. Environmental research has established that the accumulation of these nutrients in water has detrimental effects on humans and biodiversity, giving rise to carcinogenic drinking water in addition to a decrease in aquatic flora and fauna from eutrophication [9]. There is a need for a system that can continuously and accurately monitor soil conditions and the availability of macronutrients in real time. Currently, the analysis of soil nutrients takes place using the “random grid sampling” technique on portions of soil (samples) that are used for successive analyses in specialized laboratories, or for remote inspection through image acquisition [10].

Although the laboratory analyses are very accurate, they are unable to provide continuous (real-time) soil analyses. Furthermore, agricultural land is highly heterogeneous, and this factor also affects the samples taken; what is present in one sample of land may not be the same in another. Currently, this limit is counteracted by the in situ precision monitoring of soil conditions, combined with the use of Internet of Things (IoT) systems, which offer instant responses to investigations in soil conditions [11,12,13].

Most of the sensors used in laboratory analyses involve optical and/or electrochemical methods [14,15,16].

Here, an interaction approach with the soil is proposed, based on the monitoring of its electrical potentials. Electron transfer has been used by organisms for millions of years in various processes; for instances, aerobic respiration to produce adenosine triphosphate (ATP) from the hydrolysis of organic compounds through the electron transport chain in mitochondria; photosynthesis to transform radiant energy into ATP and reducing power in photosystems in the chloroplasts; or the creation of a membrane electrochemical potential that governs nutrient uptake and intracellular ionic composition.

Considering the soil, plants release considerable amounts of electrons directly to the soil in particular cases such as iron deficiency [17,18]. However, the most significant contribution of plants to generate electric power in the soil is through the photosynthates released through the roots into the soil [19]. These organic compounds are essential to sustain microbial growth and metabolism, which result in the efflux of free electrons that can be captured by different electrode configurations, namely plant-based cell (P-BC) [20], plant-microbial fuel cell (P-MFC) [21], and soil-based microbial fuel cells (MFCs) [22].

The ability to release free electrons and produce electric current varies among bacterial communities [23], and the diversity of microorganisms interacting with a plant in the soil and root microbiome is very dependent on the type of exudates released by plants. This reflects the ability of the microbiome to metabolize the exudates and impose microbiome shifts according to exudate chemical profiles [24]. The effects of plant exudates on the microbial communities must be considered, since interactions in the root microbiome, namely localization of microbes and competition, can influence electrical current generation [19]. In PMFCs, the most common bacteria belong to *Desulfobulus*, *Geobacteraceae*, *Natronocella*, *Beijerinckiaceae*, *Rhizobiales*, and *Rhodobacter*, with some species being electrochemically active [19].

Since the composition of plant exudates is quite variable, it reflects the conditions in which the plant lives, such as light, infection, predation, nutrient deficiency, flooding, drought, agrochemicals, or pollutants [25]. Changes in growth conditions will invariably affect the number and biodiversity of root-associated bacteria as well as the capacity to generate electricity from the soil. These variations can be used as markers of changes in the plant–microorganism system, and if different patterns of variation are recognized, these can be used as identifiers of the constraints that are affecting the system.

Other green options, such as solar power or biofuel, compete against other ways of using the land such as food production. Since world population is increasing, arable land is decreasing; due to global changes, alternatives to produce energy that is compatible with farming use will be revolutionary.

The design and development of two kinds of sensors that are able to capture both soil electrochemical responses and the electrical potentials generated by the interactions between soil and plants could pave the way to an innovative methodology for monitoring large areas with a network approach, thanks to these sensors’ small size and low power consumption. When connected with an IoT strategy based on an AI model in the cloud, each of these points becomes a microsystem node that interacts with the soil and sends data to a central unit in the cloud, in order to generate warning signals.

Here, the sensor design is presented, and a proof of concept of their relevance in analyzing soil samples has been tested.

## 2. Materials and Methods

Two different devices have been used: a voltammetric sensor for the electrochemical analysis of the soil, and a device for the extraction and evaluation of electrical potentials generated by interactions between soil and plants.

### 2.1. Device for Soil Voltammetric Analysis

The device used for voltammetric analysis of the soil is the BIONOTE-L, developed by the Electronics and Sensory Systems Unit of the Campus Bio-Medico University of Rome [26,27]. This device is able to operate mainly in liquids, and provides a fingerprint of samples under testing. It consists of an electronic interface and an electrochemical cell. A variable voltage (triangular wave of amplitude [−1 V; +1 V] and frequency 0.01 Hz) is generated by the interface and is applied to the electrochemical cell. These voltage inputs generate redox reactions characteristic of the analyzed solution. The electrochemical cell used in voltammetry consists of three electrodes called the working electrode (WE), reference electrode (RE), and counter electrode/auxiliary electrode (CE). The circuit diagram of the device is shown in Figure 1.

### 2.2. Device for Soil–Plant Electrical Potentials Analysis

Considering a generic fuel cell, a simplified Thevenin model (whose values depend on the operating conditions of the plant) corresponds to that in Figure 2, where V_out_ is the voltage supplied at the output of the fuel cell. V_eq_ is determined by measuring the open-circuit voltage generated by the system, identifying a parameterization that may depend on various factors, such as the number of electrodes used as well as their size, the type of soil, the climatic conditions, and the level of humidity; meanwhile, Z_eq_ is determined by measuring the current delivered to a variable load and the obtained V_out_ voltage in different operating conditions.

The next stage is to use the previously described electrical model as a reference for the design of a constant voltage-up converter (DC/DC converter), with maximum power point track (MPPT) algorithms to maximize conversion efficiency. The DC/DC converter with maximum power point track algorithm acts more like a power pump, amplifying the output voltage in relation to the input based on the system load that captures the most energy from the source [28]. This can be used to immediately charge a battery or to feed low-power systems that do not require a constant power supply. If a fixed voltage regulation is also required in conjunction with MPPT, the latter commonly acts as a pre-regulator for a second DC/DC converter with a fixed output voltage. Figure 3 depicts the block scheme of the power management system with MPPT, and regulated voltage used in the device here exploited for electrical energy capture. The electrodes used were four copper filaments (1 mm in diameter).

### 2.3. Soil and Plant Selection

The soil and plant typologies tested in this study, together with their main characteristics in terms of pH, electric conductivity, porosity, dry bulk density, and components, are reported in Table 1.

### 2.4. Measurement Protocol

The aim of this analysis was to investigate the capability of the instrument to discriminate soils with different compositions. In order to counteract possible effects due to the non-homogeneity of the sample, it was treated with the following protocol: 0.5 g of soil were taken, immersed in an aqueous solution of 10 mL of distilled water, decanted for 30 min, and finally centrifuged [29]. Then, 4 mL of supernatant were withdrawn and analyzed with the device. This detection was also performed on a sample of distilled water which was considered a reference.

Moreover, the response of the sensor to different salt concentrations was evaluated. From literature [30,31], it has been found that in a fertile soil it is possible to register a concentration of calcium chloride of 1.11 g/L, and of 3–6 g/L of NaCl.

Therefore, the solutions analyzed were the following: CaCl_2_ (1.11, 5.55, 11.10 g/L); NaCl (5, 10, 15 g/L).

## 3. Results

The results are reported with the following strategy: the first point is to demonstrate that the BIONOTE-L is able to characterize different soil samples by providing a reproducible voltammetric fingerprint specific for each different typology. Moreover, the fingerprints obtained had to be demonstrated to be specifically given by the soil composition. Thus, the soil samples were modified by adding different concentrations of two types of salt: CaCl_2_ and NaCl, in order to verify that the characteristic fingerprint registered for each different soil was modified in its shape for the presence of the added salt, and that this modification was different for each salt, specific for it, and proportional to increasing salt concentration. Finally, the interactions between plants and the soil were also verified via registration of the energy produced.

In Figure 4, the voltammograms registered of each soil sample are reported.

The three voltammograms registered for each of the four soils under testing show a typical ‘shape’ of the cyclic pattern which is common to all them because it is specific of the soil. At the same time, each of the profiles registered for each different soil type exhibit some peculiarities which should be specific to the soil typology. It is worth remarking that the reproducibility over the three samples analyzed is acceptable, yet is affected by differences in a biological sample (as the soil is) due to different sites of sampling that were tested for each measurement. ‘Acceptable reproducibility’ is intended by the authors as follows: even if the three voltammograms relative to three different samples of the same soil are not perfectly coincidental, the shape of each of them is more similar to each other than to the shapes of another threesome of voltammograms relative to another type of soil.

In order to confirm this approach and the ‘acceptable reproducibility’ of the measurements performed on the same soil typology, Figure 5 shows the scoreplot of the principal component analysis (PCA) model built on the data given by the voltammograms reported in Figure 4. Four different clusters can be observed on the scoreplot of the first two principal components (PCs), demonstrating through a multivariate approach that the voltammograms obtained for the three samples measured for each of the four types of soil are much more similar to each other than to those obtained for different types of soil.

In Figure 6, the voltammograms registered for the soil samples added with two different salts are reported. The soil typology selected for this experiment is the bonsai. The soil fingerprint registered by the BIONOTE-L is evidently altered by the presence of the salts in different ways depending on the two different salts used, and proportionally increases the detected peaks when the salt concentrations are increased.

The results observed by the graph and performances described in Figure 6 are promising: it is not just a matter of characteristic shape or profile given by a specific salt composition, which indicates the sensor’s ability to discriminate between different soils, but the BIONOTE-L also demonstrated a relevant ability to detect the salt concentration with a percentage of error that was 11% for NaCl while not competitive for CaCl_2_. It is worth noting that for the two highest concentration values, a sort of ‘saturation’ in the voltage response was observed. Those concentration values were over the normal intervals reported in literature for soil, and consequently negligible with respect to the aims of this study.

The results obtained by the device for monitoring energy produced by the interactions between plants and soil are reported in Figure 7.

## 4. Discussion

From the perspective of previous studies and from the working hypotheses, the results obtained confirm that the proposed sensors can be added to the list of the instruments used thus far (and cited in the introduction) for the characterization and monitoring of soils.

Indeed, reconsidering what was discussed in the introduction about the fact that agricultural land is highly heterogeneous, this factor also affects the samples taken; therefore, what is present on one sample of land may not be the same in another. Reproducibility is of great importance; the characteristics of the sensor tested here make it comparable with other instruments cited in the state of the art [13,14,15,16].

Its utilization on four soil typologies yielded different voltammograms characteristic for each soil kind. The reproducibility of these measurements is acceptable, since the variability of the profiles registered is mostly due to the fact that soil is not intrinsically homogeneous.

The fingerprint registered for the soil sample reflects the composition of the soil: when the soil is modified by adding different concentrations of salts, the response is specific for each salt added and proportional to its concentration in the soil. This was demonstrated with two salts, NaCl and CaCl_2_, added to bonsai soil. The PLS-DA model built on the voltammogram data has the ability to detect salt concentration with an error lower than 11% for NaCl.

It is important to compare the results obtained in this study with those from the large number of papers presenting voltammetric sensors applied to soil analyses [7,32,33,34]. Those studies have the goal of monitoring specific compounds in soil, and evidencing specific peaks; however, this is not the approach nor the main goal of the present study. The intention was not to investigate peaks and their electrochemical origins, but to study the entire shape of the voltammogram provided by the various peaks detected. This ‘holistic’ approach considered that the idea of a network of microsystems for soil monitoring is intrinsically holistic, and envisioned the use of sensors to monitor global soil modification over time; thus, we take as reference the soil itself at a certain time point, and not the presence of specific compounds. In case of anomalies, chemical analyses should be necessary and executed with standard methods in laboratories. Moreover, the current literature is mainly based on functionalized electrodes, since much of the research in the field is oriented toward identification and monitoring of specific compounds, for which a suitable functionalization can grant an effective selectivity. However, this is not the case of the BIONOTE-L, whose electrodes are not functionalized. This aspect, considering the holistic approach of the study, grants a wider selectivity spectrum and enhances reproducibility.

The BIONOTE-L device could be coupled with another device that is able to capture energy produced by the interaction between plants and soil, thus extending the microsystem node functionality when applied in a network for large-area monitoring.

Future research directions may cover the following aspects: measurements involving many different soil typologies; measurements involving more complex compositions of soil; designs of optimized electronic architectures for the microsystem node that includes the sensors tested here.

## 5. Conclusions

The tested sensor devices have been shown to be relevant in the discrimination of different soil samples; even when these samples were modified in salt concentration, the sensors were able to detect the specific salt and monitor its concentration variability. This is just a proof of concept, since the soil typologies could involve more than the four tested thus far. Furthermore, soil compositions could be more complex than the one tested here that was simplified to the evaluation of two salts. The sensors have been proven to be suitable for the realization of a microsystem node for soil monitoring via an energy mapping network, which was the aim of the study. In this regard, it is worth underlining that the application of the sensors here tested has shown good reproducibility that is comparable with technologies already used [13,14,15,16], even if lab instruments such as the ones pertaining to analytical chemistry are more performant and represent a standard. Incidentally, low-power and small-size sensors allow in-field monitoring to be organized in a sharp network. The detailed knowledge of the spatial and temporal variabilities of soil and vegetation characteristics should also make possible the optimization of cultivation through scheduling a new decision support system (DSS) to achieve the targets of Agriculture 4.0 [35]. The introduction of localized crop inspections makes it possible to split into zones each individual area, with the benefit to the organization of applying differential production systems that can reduce costs and improve productivity. In conclusion, the proof of concept here represents a preliminary result that indicates the designed and tested technology to be a good trade-off between performance and applicability in the field, without eliminating the reference standards of lab measurements [4,5,10].

Further calibration will be performed in order to fully characterize the instrumental node, implement the network, and specialize it for specific applications in the field.

## Figures and Tables

**Figure 1 micromachines-13-01440-f001:**
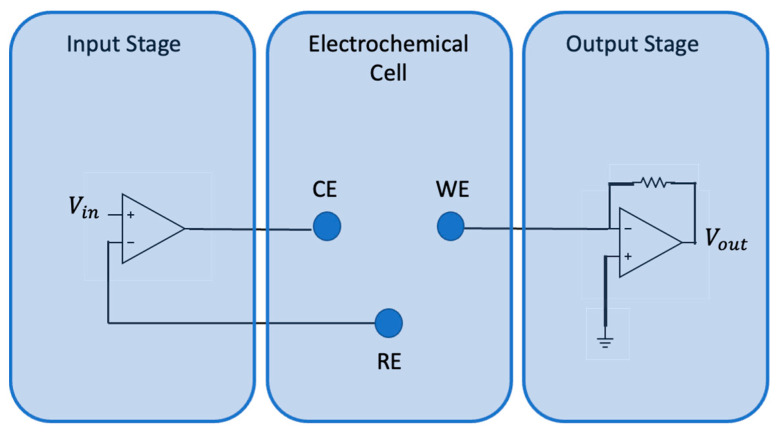
Circuit diagram of the functional blocks composing the BIONOTE-L.

**Figure 2 micromachines-13-01440-f002:**
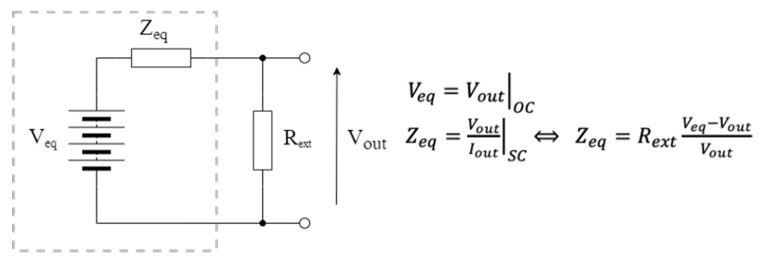
Equivalent Thevenin circuit of a simplified fuel cell model.

**Figure 3 micromachines-13-01440-f003:**
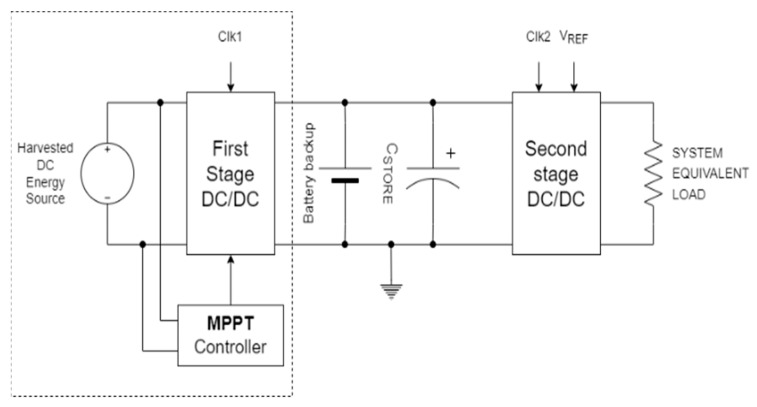
Block diagram of the device for the registration of electrical energy produced by interactions between soil and plants.

**Figure 4 micromachines-13-01440-f004:**
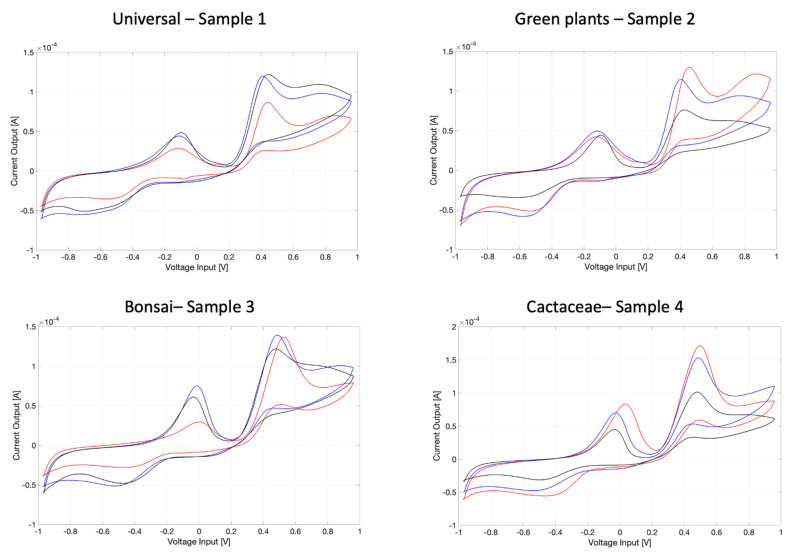
The three voltammograms (blu, red and black) registered for each of the four soils under testing: sample 1: universal; sample 2: green plants; sample 3: bonsai; sample 4: Cactaceae. The evident ‘shape’ of the voltammograms is common to all the samples because of specificities of the soil (in general); however, each of these profiles presents some peculiarities which should be specific to the soil typology. Besides, reproducibility over three samples is acceptable but of course affected by differences in a biological sample (as the soil is) due to different sites of sampling that were tested for each measurement.

**Figure 5 micromachines-13-01440-f005:**
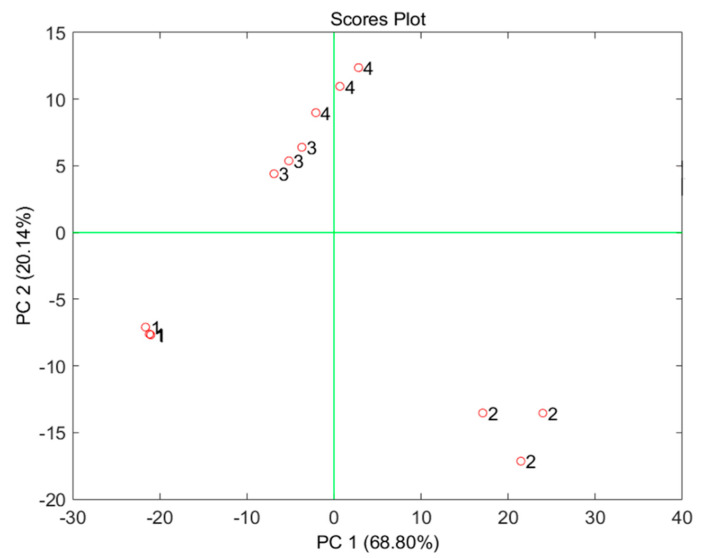
Scoreplot of the first two PCs of the PCA model built on the data reported in Figure 4. Labels: 1. universal; 2. green plants; 3. bonsai; 4. Cactacae.

**Figure 6 micromachines-13-01440-f006:**
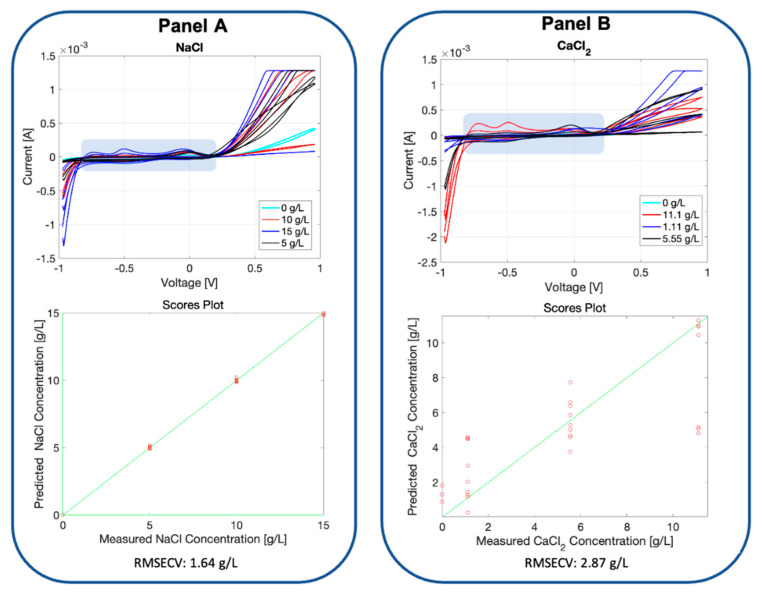
Panel A: voltammetric fingerprint registered by the BIONOTE-L in the measurement of the bonsai soil with different concentrations of NaCl (0, 5, 10, 15 g/L) added; the voltammogram is composed of 500 points. These values were used as a multidimensional data set, and were elaborated with partial least square discriminant analysis (PLS-DA), building an ad-hoc predictive model (cross-validated via the leave-one-out criterion). Its performance is described in the lower part of Panel A, reporting the predicted NaCl vs. measured NaCl concentration graph and the obtained root mean square cross-validation error (RMSECV): 1.64 g/L. Panel B: voltammetric fingerprint registered by the BIONOTE-L in the measurement of the bonsai soil with different concentrations of CaCl_2_ (0, 1.11, 5.55, 11.1 g/L) added; the voltammogram is composed of 500 points. These values were used as a multidimensional data set, and were elaborated with PLS-DA, building an ad-hoc predictive model (cross-validated via the leave-one-out criterion). Its performance is described in the lower part of Panel B, reporting the predicted CaCl_2_ vs. measured CaCl_2_ concentration graph and the obtained RMSECV: 2.87 g/L.

**Figure 7 micromachines-13-01440-f007:**
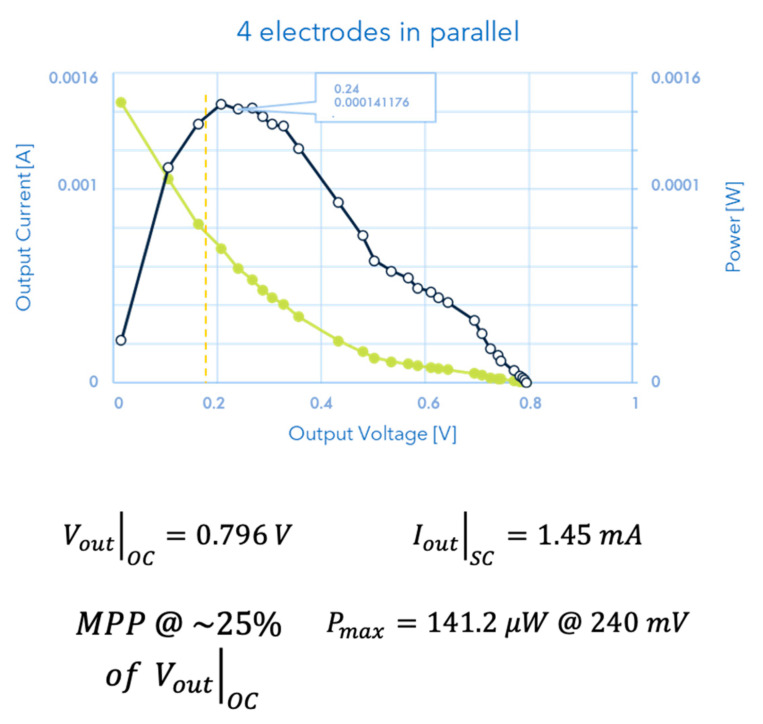
The current vs. voltage (green curve) as well as the power vs. voltage (black curve) data are reported in the graph. It is important to point out three characteristic points which show the non-neglectable values of the measured electrical quantities, both in terms of soil–plant characterization (which is the subject of the paper) and energy produced for successive harvesting or utilization: the MPP @ ~25%; the V_out_ of Open Circuit; the I_out_ of Short Circuit, whose values are put in evidence in the figure.

**Table 1 micromachines-13-01440-t001:** Soil and plant typologies. These parameters are only descriptive of the soil samples and are given by the producer. They are used to assess the different typologies as certified by the vendor.

Name	pH	Electric Conductivity [dS/m]	Porosity [Kg/m^3^]	Dry Bulk Density [%*v*/*v*]	Components
Universal	6.5	0.50	150	90	Expanded perlite (<5%)
Green plant	7	0.40	235	90	Clay
Bonsai	7	0.40	550	75	Sand and expanded clay
Cactaceae	7	0.40	600	75	Sand and expanded perlite (<5%)

## Data Availability

The data presented in this study are available on request from the corresponding author.

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
