# Peer review of "Microsystem Nodes for Soil Monitoring via an Energy Mapping Network: A Proof-of-Concept Preliminary Study"

_micromachines, 2022, doi:10.3390/mi13091440_

Round 1

Reviewer 1 Report

1.     Fig 5 A and B CV peaks seems to show over potential when using 150mg/ml in NaCl and 10mg/ml when using CaCl2, does the system is stable with this potential width? Fig 5 Calibration plot as well as other CV plots resolution needs to be improved.

2.     Comment on the sensitivity of the device.

3.  Typographical errors need to be rectified (Example: The device used for voltammetric analysis of the soil ws the BIONOTE-L).

Author Response

Thanks to the reviewer for her/his valuable comments and suggestions.

Fig 5 A and B CV peaks seems to show over potential when using 150mg/ml in NaCl and 10mg/ml when using CaCl2, does the system is stable with this potential width?

First of all authors are sorry to have reported incorrect unit measure and concentration values on figure 5. It was a formatting problem in the transition among SWs. The new figure 5 is correct, now. Reviewer’s observation about ‘over potential’ is correct. The concentration values giving those ‘saturation’ in response are in the highest part of the concentration interval selected, which is over the normal interval reported in literature, thus negligible with respect to the target of the work. This consideration has been added in the text.

 Fig 5 Calibration plot as well as other CV plots resolution needs to be improved.

The quality of Figure 5 has been improved.

  1. Comment on the sensitivity of the device.

We agree with the reviewer that sensitivity is a fundamental parameter for sensors, but it can just be applied on calibration curves. In this work the only procedure similar to a calibration has been performed in the analysis of soils with different concentrations of salts, but the aim was not test sensor ability in detecting the correct concentration, but to test the relevance of sensor response with respect to the soil composition. To this regard, the Root Mean Square Error in Cross Validation has been estimated using PLS-DA and not performing a calibration procedure for which specific voltage values and the correspondent peaks should be identified a specific salt: but this out of the scope of the work. Besides, this procedure and the consequent definition of the sensitivity could be the goal of future works devoted to key-compounds such as specific macronutrients or pollutants.

  1. Typographical errors need to be rectified (Example: The device used for voltammetric analysis of the soil ws the BIONOTE-L).

Typos have been detected and corrected.

Reviewer 2 Report

Microsystem nodes for soil monitoring via an energy mapping network: a proof of concept preliminary study Anna Sabatini , Alfiero Leoni , Gil Goncalves , Alessandro Zompanti , Marco Valerio Marchetta , Paulo Cardoso , Simone Grasso , Maria Vittoria Di Loreto , Francesco Lodato , Costanza Cenerini , Etelvina Figuera , Giorgio Pennazza * , Giuseppe Ferri , Vincenzo Stornelli , Marco Santonico

Questions and suggestions

On the base of the phrase "Here the sensor design is presented and a proof of concept of their relevance in analyzing soil samples has been tested" (lines 131-132), the referee asks the following aspects.

1. Table 1. Authors report the "Name", "pH", "Electric conductivity", "Porosity", "dry bulk density" and "Components". What do the parameters in table 1 are crucial or strongly influence the detections performed by the authors, on the soil samples chosen? It's not clear if some of these parameters are or aren't determinant in the soil samples detection described by the authors. On the other hand, if authors made measurements with the sensor proposed by them (see lines from 138 to 147), the referee's question is, are these parameters only descriptive or there are some relationship with results obtained?

2. Figure 4. According the referee there is something no clear. Soil samples are four and voltammograms are four, according the authors. Where is the fourth voltammogram? If the referee sees only three per soil sample? Then, assuming that voltammograms are three, the question is, which is the correspondence of the three CVs reported? What does every CV signal mean, per every soil sample? Caption must be revise and rewrite, and must be in agreement with the description reported on the manuscript. The caption must be improved.

3. In figure 5 authors affirm that the voltammograms reported are related to three salts. One is NaCl, the other one is CaCl2, which is the third one? Does the referee miss or misunderstanding something?

4. Figure 6. Why f(x) values are not reported with scientific numbers? Besides, the referee supposes that the dark-blue curve is related to the Power [W], and the light-green line is assigned to output current [A] , does not? Caption is unclear and in disagreement with the description on the text.

5. Conclusions are poor and not well supported by results, that at the same time are not well described, even included conditions for the CV signals recorded.

The referee suggests a carefully and critical revision of the proposed manuscript, and then to submit it again. 

Author Response

Thanks to the reviewer for her/his valuable comments and suggestions.

On the base of the phrase "Here the sensor design is presented and a proof of concept of their relevance in analyzing soil samples has been tested" (lines 131-132), the referee asks the following aspects.

  1. Table 1. Authors report the "Name", "pH", "Electric conductivity", "Porosity", "dry bulk density" and "Components". What do the parameters in table 1 are crucial or strongly influence the detections performed by the authors, on the soil samples chosen? It's not clear if some of these parameters are or aren't determinant in the soil samples detection described by the authors. On the other hand, if authors made measurements with the sensor proposed by them (see lines from 138 to 147), the referee's question is, are these parameters only descriptive or there are some relationship with results obtained?

These parameters are only descriptive of the soil samples and are given by the producer. They are used to assess the different typology as certified by the vendor. This information has been added in the text.

  1. Figure 4. According the referee there is something no clear. Soil samples are four and voltammograms are four, according the authors. Where is the fourth voltammogram? If the referee sees only three per soil sample? Then, assuming that voltammograms are three, the question is, which is the correspondence of the three CVs reported? What does every CV signal mean, per every soil sample? Caption must be revise and rewrite, and must be in agreement with the description reported on the manuscript. The caption must be improved.

Authors agree with the reviewer: there is a typo in the caption and in the text. Figure 4 has formed up by 4 panels. Each panel is referred to a soil typology (as specified in the title upon each panel and in the caption). In each panel 3 voltammograms are reported (very similar indeed) relative to three different samples of the same soil. Thanks, the caption and the text have been revised.

  1. In figure 5 authors affirm that the voltammograms reported are related to three salts. One is NaCl, the other one is CaCl2, which is the third one? Does the referee miss or misunderstanding something?

Authors are very sorry. The salts are two and the reference to a third salt is a typo.

  1. Figure 6. Why f(x) values are not reported with scientific numbers? Besides, the referee supposes that the dark-blue curve is related to the Power [W], and the light-green line is assigned to output current [A] , does not? Caption is unclear and in disagreement with the description on the text.

Thanks, reviewer’s comment is correct: the caption has been improved.

  1. Conclusions are poor and not well supported by results, that at the same time are not well described, even included conditions for the CV signals recorded.

The referee suggests a carefully and critical revision of the proposed manuscript, and then to submit it again. 

The work has been carefully and critically revised following reviewer’s suggestion. Authors also hope that the correction in the figures has contributed to better clarify the results obtained.

Reviewer 3 Report

The paper manuscript “Microsystem nodes for soil monitoring via an energy mapping network: a proof of concept preliminary study.”

Overall, this is a well written manuscript and considred adequate to be accepted.

Nevertheless, the authors should revise better their Abstract, Introduction and Discussion. Please shorten the big phrases. Furthermore, the study must clearly examine how the findings relate to previous research in this area.

Minor comments follow.

1)Please check abbreviations with consistency in main text. Define it at the first appearance, then use it after the definition (e.g. Sustainable agriculture (SA) and  precision agriculture (PA) (repeated many times; could be defined) , ATP (not defined) AI model (not defined), DC/DC converter (not defined) etc.).

2) Line31: Change “CaCl2” to ” CaCl2”.

3) Line 68: Change “physiologyto” to .” physiology to”.

4) Line 80: Change “analyzes” to ” analyses”.

5) Line 115: Check space(s).

6) Line 191:  Please revise the sentence:”…concentration of calcium chloride of 1.11 g / l and of e and 3-6 g / l of NaCl.” Several typos seems to be the case.

7) Line 214: Change “is common to all tehm” to .” is common to all them”.

8) Line 222: Change “eveidently” to “evidently”.

9) Please specify/provide further information about the 4 soil types used, if possible.

10) In lines  199-200 it is noted: Thus the soil samples have been modified by adding different concentration of three types of salt: K, CaCl2  and NaCl”

Please specify which salt corresponds to K. However, no results are available for a third salt (e.g see figure 5). Please clarify and revise the manuscript where needed.

11)Please add a reference.  “……Decision Support System (DSS) to achieve the target of the Agriculture 4.0”.

I will be glad to provide further details if needed and thank you for contacting me.

Author Response

The paper manuscript “Microsystem nodes for soil monitoring via an energy mapping network: a proof of concept preliminary study.”

Overall, this is a well written manuscript and considred adequate to be accepted.

Thanks to the reviewer for her/his kind appreciation.

Nevertheless, the authors should revise better their Abstract, Introduction and Discussion. Please shorten the big phrases. Furthermore, the study must clearly examine how the findings relate to previous research in this area.

These aspects have been taken into account and manuscript’s quality has been consequently improved.

Minor comments follow.

1)Please check abbreviations with consistency in main text. Define it at the first appearance, then use it after the definition (e.g. Sustainable agriculture (SA) and  precision agriculture (PA) (repeated many times; could be defined) , ATP (not defined) AI model (not defined), DC/DC converter (not defined) etc.).

Corrected, thanks.

2) Line31: Change “CaCl2” to ” CaCl2”.

Changed, thanks.

3) Line 68: Change “physiologyto” to .” physiology to”.

Changed, thanks.

4) Line 80: Change “analyzes” to ” analyses”.

Changed, thanks.

5) Line 115: Check space(s).

Checked and corrected, thanks.

6) Line 191:  Please revise the sentence:”…concentration of calcium chloride of 1.11 g / l and of e and 3-6 g / l of NaCl.” Several typos seems to be the case.

Revised, thanks.

7) Line 214: Change “is common to all tehm” to .” is common to all them”.

Corrected, thanks.

8) Line 222: Change “eveidently” to “evidently”.

Changed, thanks.

9) Please specify/provide further information about the 4 soil types used, if possible.

These parameters are only descriptive of the soil samples and are given by the producer. They are used to assess the different typology as certified by the vendor. This information has been added in the text. We do not have any other data to report.

10) In lines  199-200 it is noted: Thus the soil samples have been modified by adding different concentration of three types of salt: K, CaCl2  and NaCl”

Please specify which salt corresponds to K. However, no results are available for a third salt (e.g see figure 5). Please clarify and revise the manuscript where needed.

Authors are very sorry. The salts are two and the reference to a third salt (K)  is a typos.

11)Please add a reference.  “……Decision Support System (DSS) to achieve the target of the Agriculture 4.0”.

 Ok, added.

Round 2

Reviewer 2 Report

I read the new version, and some improvement have been made of course. But the impression I have is that they don't say or explain well all information.
For example, they say that have four soil and three voltammograms per soil, all of them reproducible. What does that mean?
Also me can public something that looks like "reproducible" but what does it mean? If you don't describe peaks, the chemistry behinds this peak (briefly of course) and eventually report some reference on matter, what are you saying new?
People that have publish on electrochemical approach on soils there are a lot around the world!
I understand that is a prof of concept about a voltametri sensor, but if authors doesn't explain the results clearly this is not acceptable for me.
Authors made a good work, they should explain well it.

Authors should implement information on the Electrochemical results.
Describe results obtained with voltammetric sensor is mandatory.
The peaks considering only electrochemical imprinting, without any description about what this signals mean is not enough.
They should improve the description of voltammograms and check literature on matter.

Author Response

Responses to reviewer 2

I read the new version, and some improvement have been made of course.

Thanks to the reviewer for her/his appreciation.

But the impression I have is that they don't say or explain well all information.

We understand the point: following reviewer’s suggestions authors have added important details.

For example, they say that have four soil and three voltammograms per soil, all of them reproducible. What does that mean? Also me can public something that looks like "reproducible" but what does it mean?

We agree with the reviewer that the term ‘acceptable reproducibility’ has not been clearly explained, but just reported as implicit meaning, and this is not enough clear. In the text reported from line 219 to line 225 authors say:

The three voltammograms registered for each of the 4 soils under test shows to have a ‘shape’ of the cyclic pattern which is common to all them because it is specific of the soil. At the same time, each of the profiles registered for each different soil type, puts in evidence some peculiarities which should be specific of the soil typology. It is worth remarking that the reproducibility over the three samples analysed is acceptable but of course it is affected by the difference in a biological sample (as the soil is) due to the different site of sampling tested for each measurement.”

For authors, in the context of the experiment here presented, ‘acceptable reproducibility’ means that, even if the three voltammograms relative to three different samples of the same soil are not perfectly coincident, the shape of each of them is more similar one to each other than to the shape of another threesome of voltammograms relative to another type of soil. This explanation  has been added in the revised paper.

In order to confirm this approach and the ‘acceptable reproducibility’ of the measurements performed on the same soil typology, authors have reported in fig. 5 the scoreplot of the Principal Component Analysis model built on the data given by the voltammograms reported in fig. 4. Four different clusters can be observed on the scoreplot of the first two PCs, demonstrating with a multivariate approach that the voltammograms obtained for the three samples measured for each of the four types of soil are much more similar to each other than those obtained for the different types of soil.

If you don't describe peaks, the chemistry behinds this peak (briefly of course) and eventually report some reference on matter, what are you saying new? People that have publish on electrochemical approach on soils there are a lot around the world!

Authors thanks the reviewer for the opportunity to better describe the approach of the work. Of course, there are many papers describing the use of electrochemical sensors for soil analysis: authors cited some of them and others have been added. Authors agree with the reviewer: it is important to cite some more papers, because, doing this, the novelty of the work can emerge. All the works cited have the goal of monitoring specific compounds in soil, evidencing specific peaks: this is not the approach of the present work, which is not intended to study peaks and their electrochemical origin, but the whole shape of the voltammogram given by the various peaks detected. Of course, this approach is ‘holistic’, but the idea of a network of microsystems for soil monitoring is intrinsically holistic and envisioned for monitoring global soil modification over time, thus taking as reference the soil itself at a certain time-point, and not the presence of specific compounds. In case of anomaly, chemical analyses should be necessary and executed with standard methods in lab. Thus, the sensor network should serve as warning system. This concept has been added in the discussion. To authors’ best knowledge, literature on voltammetry applied to soil analysis (of which the papers cited are just representative) is oriented to specific compounds identification and they are very often based on functionalized electrodes, while the screen printed electrode here used are not functionalized.

I understand that is a prof of concept about a voltametri sensor, but if authors doesn't explain the results clearly this is not acceptable for me. Authors made a good work, they should explain well it.

Thanks to the reviewer. Indeed, the way authors can better explain the work is to better specify the envisioned utilization of the sensor network, also giving support with multivariate data analysis, as said in the previous responses.

Authors should implement information on the Electrochemical results.
Describe results obtained with voltammetric sensor is mandatory.
The peaks considering only electrochemical imprinting, without any description about what this signals mean is not enough. They should improve the description of voltammograms and check literature on matter.

As already said in the previous responses, it is true there are many but they look for specific compounds and they are functionalized. The BIONOTE-L uses non-functionalized screen-printed electrodes and the objective of the proof-of-concept is not to identify specific compounds. Moreover, we did not perform the chemical analysis of our soils, but that was not the purpose of the work.

The meaning of the signal (by an electrochemical point of view) is not applicable without the chemical analysis of the soil, which is out of the scope of the work. Authors understand reviewer’s perplexity on this, and this is why we decided to apply an inverse procedure, by adding certain salt concentrations to the soil under analysis, in order to clarify the pertinence of the signal with respect to its composition (see lines 293-297) and to complement the presentation of the results with multivariate models.

Round 3

Reviewer 2 Report

The manuscript has been improved, for that reason it can be accepted and published. The referee recommends a careful final reading of the proposed manuscript to verify that the results contained therein are satisfactorily explained.